Inhibition of excessive mitophagy by N-acetyl-L-tryptophan confers hepatoprotection against Ischemia-Reperfusion injury in rats

Li Huiting 1
Pan Yitong 1
Wu Hongjuan 2
Yu Shuna 1
Wang Jianxin 1
Zheng Jie 3
Wang Can 1
Li Jianguo ljg7111@wfmc.edu.cn 1
Jiang Jiying jiangjy@wfmc.edu.cn 1
1 Department of Anatomy, Weifang Medical University , Weifang , China
2 Morphology Lab, Weifang Medical University , Weifang , China
3 Department of Pathology, Weifang Medical University , Weifang , China
Naz Sarwat
Electronic publication date: 2020 Apr 9
Publication date: 2020
Volume: 8
Electronic Location ID: e8665
Received 2019 Oct 8; Accepted 2020 Jan 29
Copyright: ©2020 Li et al.
Copyright year: 2020
Copyright holder: Li et al.
License: This is an open access article distributed under the terms of the Creative Commons Attribution License, which permits unrestricted use, distribution, reproduction and adaptation in any medium and for any purpose provided that it is properly attributed. For attribution, the original author(s), title, publication source (PeerJ) and either DOI or URL of the article must be cited.
License URL: https://creativecommons.org/licenses/by/4.0/

Keywords: Hepatic ischemia-reperfusion injury, H2O2, N-acetyl-L-tryptophan, Mitophagy

Funding: National Science Foundation of China 81760567 81802474 Key R&D Program of Shandong Province GG201809200094 Natural Science Foundation of Shandong Province, China ZR2018MC012 ZR2014HL020 ZR2014HL021 Project Funding for the Development of Medical and Health Science and Technology in Shandong Province 2014WS0464 Neurologic Disorders and Regenerative Repair Lab “13th five-year plan” Key Lab of Shandong Higher Education The study was supported by the National Science Foundation of China (Grant No. 81760567; 81802474), the Key R&D Program of Shandong Province (GG201809200094), the Natural Science Foundation of Shandong Province, China (ZR2018MC012; ZR2014HL020; ZR2014HL021) and the Project Funding for the Development of Medical and Health Science and Technology in Shandong Province (2014WS0464) and the Neurologic Disorders and Regenerative Repair Lab “13th five-year plan” Key Lab of Shandong Higher Education. The funders had no role in study design, data collection and analysis, decision to publish, or preparation of the manuscript.

==============================
In order to investigate the mechnism of hepatoprotective of N-acetyl-L-tryptophan (L-NAT) against ischemia-reperfusion (I/R) injury, the effects of L-NAT were investigated in hepatic ischemia-reperfusion injury (HIRI) models both in vitro and in vivo, which were made by BRL cells and Sprague-Dawley (SD) rats, respectively. The cell viability of hepatocyte was assessed by cell counting kit-8 (CCK-8) staining. The activation of autophagy was detected by electron microscopy (EM), quantitative real-time PCR (qRT-PCR), Western blotting and immunofluorescence. The activation of mitophagy was determined by the change of autophagy related protein, change of mitochondrial structure and function, co-location of autophagy protein and MitoTracker. Results showed that the morphological structures of hepatocytes were changed significantly after HIRI, and the cell viability of hydrogen peroxide (H2O2)-induced BRL cells was decreased. Autophagy markers Beclin1, microtubule associated protein 1 light chain 3-II (LC3-II) and autophagy related protein-7 (ATG-7) were highly expressed and the expression of SQSTM1 (P62) was decreased after HIRI, which suggested that autophagy of hepatocytes was activated after I/R. The reduction of ATP, mitochondrial DNA (mtDNA) and the mitochondrial transmembrane potential (ΔΨm) after H2O2-induced revealed that function of mitochondrial had also undergone significant changes. The increased expression of autophagy protein, destructure of mitochondria and mitochondrial dysfunction, the increased co-location of Beclin1 and MitoTracker induced by H2O2 implied the excessive mitophagy. The expression of the autophagy protein was increased by 3-Methyladenine (3-MA), providing another piece of evidence. Importantly, all changes were restored by L-NAT pretreament. In conclusion, the present findings demonstrate that excessive mitophagy involved in the process of HIRI and L-NAT may protect hepatocytes against HIRI by inhibiting activation of mitophagy and improving the structure and function of mitochondria.

Introduction

Hepatic ischemia-reperfusion injury (HIRI) occurs in many clinical situations, including hepatic trauma, hypoperfusion due to vascular obstruction or hypovolemic shock, liver transplantation, and liver tumors resection (Selzner, Rudiger & Clavien, 2003). Ischemia-reperfusion (I/R) injury during liver surgery and liver transplantation is regarded as the primary cause of liver failure. HIRI is also a source of major complications in clinical practice affecting perioperative morbidity, mortality, and recovery (Park et al., 2009; Kalimeris et al., 2016). Although many different therapeutic interventions have been explored, there are still no effective treatments to prevent HIRI. Therefore, it is necessary to explore effective therapeutic interventions in HIRI.

During the process of HIRI, the elevated reactive oxygen species (ROS) level and subsequently depletion of endogenous antioxidants result in total breakdown of the endogenous antioxidant defense and the failure of protecting the hepatocytes from oxidative damage (Ji et al., 2012; Lin et al., 2016; Shadel & Horvath, 2015). Numerous evidences indicate that mitochondria are main source of ROS, and extremely susceptible to oxidative damage (Singh et al., 2019). Moreover, ROS mediated mitochondrial oxidative damage compel to more production of ROS. The elevated ROS can either directly or indirectly lead to mitochondrial dysfunction. Therefore, removal of damaged mitochondria by mitophagy is essential for the physiological function of cells. Previous studies have shown that moderate autophagy can alleviate I/R injury in heart, brain and renal by eliminating damaged mitochondria reducing oxidative stress and free radical production (Li, Chen & Gibson, 2013; Ling et al., 2011; Rautou et al., 2010), but excessive eliminating mitochondria could cause cell death, namely atophagic cell death. Amadoro et al. (2014) have demonstrated that excessive mitophagy was involved in many disease process, such as neurodegenerative diseases immunological diseases, and cancers. Therefore, we speculated that excessive mitophagy may be a new therapeutic target for HIRI.

N-acetyl-L-tryptophan (L-NAT), a potent scavenger of ROS and inhibitor of cytochrome c release from mitochondria, is reported to provide neuroprotection in neurodegenerative diseases (Li et al., 2015), such as reducing the damage of Parkinson’s disease (Thornton & Vink, 2012) and alleviating brain edema and axonal injury in traumatic brain injury and stroke (Vink et al., 2004; Donkin et al., 2009; Turner, Blumbergs & Vink, 2005). Another study also showed that L-NAT, an antagonist of the neurokinin 1 receptor (NK-1R) known to disrupt the binding of Substance P (SP) to NK-1R, played a protective role on neurodegenerative diseases (Sirianni et al., 2015). Our previous studies indicated that L-NAT could not only reduce the damage of the primary hippocampal neurons and NSC-34 cells induced by hydrogen peroxide (H2O2) in vitro, but also alleviate ALS and cerebral ischemia/hypoxia injury in mouse model in vivo (Codogno & Meijer, 2013; Jiang et al., 2014). Our recent work confirmed that L-NAT has hepatoprotective effects in vitro and vivo through inhibiting the disruption of hepatocytes, improving the cell viability, attenuating the inflammation and the expression of RIP2, Caspase-1 and IL-1β (Wang et al., 2019). However, the relationship between mitophagy and the hepatoprotective of L-NAT are not fully understood. In this study, we investigated the effects of L-NAT on hepatocytes morphology, the structure and function of mitochondria, and activation of autophagy during the period of HIRI, which may provide experimental evidence for application of L-NAT on HIRI.

Material and Methods

Chemicals

L-NAT, Beclin1, microtubule- associated protein 1 light chain 3-II (LC3-II), autophagy related protein 7 (ATG-7), SQSTM1 (P62) antibodies and 3-Methyladenine (3-MA) were purchased from Sigma-Aldrich (St.Louis, MO, USA) and GAPDH antibody was obtained from Proteintech Group (Chicago, USA). Secondary anti-rabbit antibody was purchased from Amersham Pharmacia Biotech (Piscataway, NJ). The enhanced chemiluminescence (ECL) system was obtained from Amersham Pharmacia Biotech (Piscataway, NJ). RIPA lysis was purchased from Solarbio (Beijing, China). The cell counting kit-8 (CCK-8) was purchased from 7sea-Biotech (Shanghai, China). ATP assay kit was purchased from Beyotime Biotechnology (Shanghai, China) and MitoTracker Red kit came from Yeasen Biotech (Shanghai, China) and DAPI came from Life Technologies.

Animals

Healthy male Sprague-Dawley (SD) rats weighing 200-220 g were purchased from Pengyue experimental animal center in Jinan, China (Weifang Medical University Medical Ethics Committee provided full approval for this research (No. 2017253)). They were randomly divided into sham group, I/R group, and I/R + L-NAT group, with 6 rats in each group. Rats were fasted for 12 h before surgery, and were given free access to water. In I/R + L-NAT group, L-NAT (10 mg/kg) was intraperitoneal injected 30 min before modeling (Wang et al., 2019). The rats were anesthetized by intraperitoneal injection of ketamine, the left and middle branches of the hepatic pedicle were occluded with a non-traumatic vascular clamp in I/R + L-NAT group and I/R group. The success of the model was apparent once the liver color turned from red to dark purple. After 45 min of ischemia, the clip was removed to allow hepatic reperfusion. In sham group, the rats underwent the same surgery but no vessel clamps were placed. According to the previous literature of our laboratory, the most obvious liver function damage was after 6 h of reperfusion, so we took the liver tissue after 6 h of reperfusion. The Animal Ethics Committee of the University approved all working protocols.

Cell culture and treatment

The rat hepatocyte BRL cell line was purchased from the Chinese Academy of Sciences Cell Bank (Shanghai, China). BRL cells were cultured in a 37 °C incubator which is a humidified environment of 95% air/5% CO2. The cells were divided into three groups: control group, H2O2 group and H2O2+ L-NAT group. Referring to the previous literature in our laboratory, oxidative damage model of BRL cells were prepared by pre-treatment with 200 µM H2O2 for 6 h. And H2O2+ L-NAT group was pre-treated with 10 µM L-NAT for 2 h before adding H2O2 (Wang et al., 2019).

Cell viability assay

Cell viability was determined using CCK-8 colorimetric kit. BRL cells were adjusted as 2 × 105 cells/mL, then were inoculated in 96-well plate with 100 µL/well and were cultured overnight, when a partial monolayer was formed. Then, cells were exposed to 200 µM H2O2 with or without L-NAT for 6 h. Subsequently, cells were cultured in a solution of 100 µL CCK-8/mL DMEM for 1 h in a CO2 incubator in darkness. The absorbance of the sample was measured at 450 nm using a microplate reader (Thermo, Massachusetts, USA). The experiments were performed in 5 replicate holes and repeated three times.

Enhanced ATP assay

The level of intracellular ATP was determined according to the instructions using an ATP assay kit. A same amount of BRL cells in 6-well plates were lysed and centrifuged at 12,000 g for 5 min at 4 °C. Subsequently, 20 µL of the supernatant was mixed with 100 µL of luciferase reagent in an opaque 96-well plate and was measured using the chemiluminometer.

Transmission election microscopy (TEM)

The perfused liver tissue was fixed in PBS (pH = 7.4) containing 2.5% glutaraldehyde for at least 2 h, which was then fixed in 1% osmium tetroxide for 1 h at 4 °C temperature. Tissues were embedded in an epoxy resin and made into 100 nm ultrathin slices. The ultrastructure of the sample was observed using TEM (JEM. 1010, JEOL, Tokyo, Japan).

Immunofluorescence technique

Liver tissue was perfused and fixed, and 10-µm-thick frozen sections were prepared, fixed in 4% paraformaldehyde for 15 min, and washed with PBS. The endogenous peroxidase was inactivated by blocking with 1% goat serum for 30 min at room temperature and then incubated overnight at 4 °C with diluted primary antibody (LC3-II, 1:200, Sigma; Beclin1, 1:200, Sigma; ATG-7, 1:200, Sigma; P62, 1:200, Sigma). The following day, it was washed three times with PBS, incubated with the secondary antibody for 60 min at room temperature, washed three times with PBS, and nucleus were stained by DAPI (1 mg/mL), then observed under the fluorescence microscope (Olympus, Japan).

MitoTracker staining mitochondria

After seeding cells on coverslips and treating with the drug, according to the instructions of MitoTracker Red CMXRos kit (Yeasen, Shanghai, China), 200 nM H-DMEM medium was added onto each slide, and then incubated in 37 °C incubator for 30 min. Cells were washed with PBS, and fixed with paraformaldehyde and permeabilized with cold acetone, then observed cells under the fluorescence microscope (Olympus, Japan).

Rhodamine 123 staining

BRL cells were cultured on slide and pretreated with 10 µM L-NAT for 2 h and then treated with 200 µM H2O2 for 6 h. Mitochondrial membrane potential was determined using rhodamine 123, a kind of lipophilic cationic fluorescent dye that can pass through the cell membrane of live cells for detecting mitochondrial membrane potential (Jiang et al., 2014). Briefly, the processed cells were directly incubated with 2 µM rhodamine 123 for 40 min in 37 °C CO2 incubator in dark, followed by rinsing with PBS, and stained nucleus with DAPI (1 mg/mL). Images were taken using a fluorescence microscope.

Flow cytometry

BRL cells were adjusted to 2 × 105 cells/mL, then were inoculated in 6-well plate with 1 mL/well and cultured overnight. After treatment with L-NAT and H2O2, then cells were collected into centrifuge tubes and centrifuged at 2,000 rpm for 5 min, the supernatant was discarded and PBS was added. Rhodamine 123 was added to the suspension at a concentration of 2 µg/mL, incubated at 37 °C for 40 min, and then centrifuged at 2,000 rpm for 5 min. Cells were washed twice with PBS, and detected by flow cytometry. The excitation wavelength was 507 nm, and the emission wavelength was 530 nm.

Quantitative real-time PCR (qRT-PCR)

Total RNA was extracted from the treated liver tissue and BRL cells according to the instructions of TRIzol agentia (Life, USA), to reverse transcripted in cDNA and PCR amplification was performed with 20 µL reaction system (Wang et al., 2019).

Western blotting

Tissue or cells were exposed to the mixture of RIPA lysis buffer and phenylmethylsulfonyl (PMSF) fluoride, lysed on ice, and centrifuged at 12,000 g for 15 min at 4 °C to extract total protein and protein concentration was determined by BCA method. Proteins were separated by electrophoresis and then transferred onto a PVDF membrane by electro-blotting. Samples were incubated overnight at 4 °C with antibodies such as LC3-II (1:500), Beclin1 (1:500), ATG-7 (1:200), P62 (1:500) and GAPDH (1:1000). The images were analyzed by ImageJ software. The results were normalized to the ratio of target protein optical density values to those of the internal reference protein GAPDH.

Statistical analysis

All experiments were repeated three times and data were presented as the mean ± SD values. Statistical analyses of the data were performed using a one-way analysis of variance (ANOVA) followed by the least significant difference (LSD) test with Statistical Package for the Social Sciences (SPSS, version 22.0) software, p < 0.05 were considered statistically significant.

Results

Effect of L-NAT on autophagy after hepatic ischemia-reperfusion injury

Given the evidence that excessive autophagy related to hepatocyte injury during HIRI, and the modulation of autophagy may render hepatocyte resistant to I/R injury. Subsequently, the change of autophagy activation was investigated. The times sequential changes of Beclin1 was investigated by qRT-PCR and Western blotting during 0 h, 1 h, 4 h, 6 h, 12 h and 24 h of reperfusion, as shown in Figs. 1A–1C, both the mRNA and protein expression of Beclin1 peaked at 6 h after reperfusion. Accordingly, rats were subjected to 45 min of hepatic ischemia followed by reperfusion 6 h for subsequent experiments.

Figure 1 The time-series expression of Beclin1 mRNA and protein during HIRI.

(A) The time-series expression of Beclin1 determined by qRT-PCR. (B–C) The time-series expression of Beclin1 determined by Western blotting. The data are the mean ± SD (n = 3), *p < 0.05, **p < 0.01, ***p < 0.001 compared with the related sham group.

To understand whether L-NAT could reduce autophagy activation, we next investigated the expression of autophagy markers (LC3-II, Beclin1, ATG-7 and P62) after HIRI present with or without L-NAT using the Rat HIRI model and H2O2-induced oxidative damage. Results of qRT-PCR and Western blotting showed that the expression levels of LC3-II, Beclin1, and ATG-7 in I/R group showed an increase than those in sham group, while the P62 level dropped to 0.6-fold, and L-NAT administration could reverse those changes of autophagy markers (Figs. 2A–2I). Immunofluorescence assays showed that LC3-II, Beclin1, and ATG-7 staining (green fluorescence) increased, P62 staining (green fluorescence) significantly decreased in I/R group compared with the sham group (Fig. 2J). As shown in Figs. 3A–3J, results in vitro are consistent with that in vivo. These data indicated that L-NAT could regulate the expression of autophagic protein during HIRI injury.

Figure 2 Changes of autophagy marker after HIRI in vivo.

(A–I) Relative mRNA and Western blotting analysis of autophagy markers (Beclin1, LC3-II, ATG-7 and P62) in rat liver tissues (n = 3). (J–U) Immunofluorescence staining (200×) in rat liver tissues, bar = 50 µm. The data are the mean ± SD, *p < 0.05, **p < 0.01, ***p < 0.001 compared with the related sham group, #p < 0.05, ##p < 0.01, ###p < 0.001 compared with the related I/R group.

Figure 3 The changes of autophagy protein in H2O2-induced BRL cells in vitro.

(A–I) Relative mRNA and Western blotting analysis of autophagy markers (Beclin1, LC3-II, ATG-7 and P62) in BRL cells (n = 3). (J–U) Immunofluorescence staining (200×) in BRL cells, bar = 50 µm. The data are the mean ± SD, *p < 0.05, **p < 0.01, ***p < 0.001 compared with the related control group, #p < 0.05, ##p < 0.01, ###p < 0.001 compared with related H2O2 group.

To confirm the protective effect of L-NAT on HIRI-induced autophagy, the autophagosome in hepatocyte were observed by EM. Results showed that most of autophagosomes in I/R group, showed monolayer membrance, wrapped the degraded mitochondria and other cytosolic component. Meanwhile, in I/R group, autophagosome volume is increased approximately 1.5–2 times larger than that in sham group. And all those morphological changes of autophagosomes were reversed by L-NAT pretreated (Fig. 4). Those data demonstrated that L-NAT could decrease the formation of autophagosome.

Figure 4 Changes of autophagosomes after HIRI detected by transmission electron microscopy (10,000×).

(A–C) The arrows indicate autophagosomes, bar = 1 µm.

Effects of L-NAT on mitochondrial structure

A large body of evidence indicates that mitochondrial dysfunction plays a key role in I/R injury in heart, brain, liver, kidney and other organs. To investigate whether the effect of L-NAT on I/R is related to mitochondrial protection, we investigated the effect of L-NAT on mitochondrial morphology. EM analysis showed that mitochondria of hepatocyte in sham group were evenly distributed in the cytoplasm, with well-arranged cristae structure, while the number of mitochondria in I/R group increased and was centralized distribution around the nucleus, the mitochondrial cristaes were disordered, while the damaged mitochondria were phagocytosed by the vacuolar structure. L-NAT pretreatment improved morphological changes of mitochondria induced by I/R injury (Fig. 5).

Figure 5 Changes of mitochondria after HIRI detected by transmission electron microscopy (10,000×).

(A–C) The arrows indicate damaged mitochondria which were phagocytosed by the vacuolar, bar = 1 µm.

Effects of L-NAT on mitochondrial function

ATP, produced by mitochondria, is the main form of intracellular energy supply. To investigate whether L-NAT improves mitochondrial function changes caused by HIRI, we measured the ATP content in H2O2-induced oxidative damage model with or without L-NAT. Results showed that compared with control group, H2O2-mediated BRL cells caused mitochondrial damage, resulting in a ATP decrease of 0.35-fold, which was reversed to 0.8-fold by L-NAT pretreatment. Those changes of ATP indicated that L-NAT can rescue mitochondrial function (Fig. 6A).

Figure 6 Effects of L-NAT on mitochondrial function after HIRI.

(A–D) ATP contention, the ratio of mtAtp6/Rp113 mRNA, the expression of ND1 mRNA and COX-1 mRNA. (E–K) Using rhodamine123 staining, mitochondrial membrane potential in BRL cells was detected by flow cytometry (E–H) and fluorescence microscope (I–K), bar = 50 µm. The data are the mean ± SD, *p < 0.05, **p < 0.01, ***p < 0.001 compared with the related sham group, #p < 0.05, ##p < 0.01 compared with the related I/R group.

The above results showed that L-NAT can protect mitochondrial structure and ATP which is caused by I/R-induced, thereby alleviating mitochondrial dysfunction. A large body of evidence indicated that thirteen kinds of proteins encoded by mitochondrial DNA (mtDNA) are closely related to mitochondrial function, and maintaining stability of the quantity and quality of mtDNA are critical for the maintenance of mitochondrial function. In order to clarify the effect of L-NAT on mtDNA changes after I/R, we designed primers for the coding genes ND1 and COX-1 of mtDNA heavy and light chains to detect copy number of mtDNA. Additionally mtAtp6 is used to represent mtDNA (Codogno & Meijer, 2013), and expression of Rp113 is used to represent cellular total DNA, therefore, the ratio of them represents the content of mitochondria in each cell. Subsequently, qRT-PCR was used to detect the expression of the ND1, COX-1, mtAtp6 and Rp113 in I/R-mediated oxidative stress injury model. We found that the radio of mtAtp6/Rp113 was increased, which is 1.42-fold higher than that in sham group, and the expression of ND1 and COX-1 was attenuated to 0.67-fold and 0.74-fold in I/R group compared with sham group, respectively. The L-NAT pre-treatment group significantly improved these changes (Figs. 6B–6D). Taken together, the experimental results provided strong evidence that maintaining the quantity and quality of mtDNA stability might be closely related to the protective effect of L-NAT.

As an extension of the above results, the mitochondrial transmembrane potential (ΔΨm) assay was used as a specific test for the earliest events of mitochondrial injury. The results showed that rhodamine 123 fluorescence in control group cells showed a punctate distribution, which has an appearance of a grainy rhodamine 123 fluorescence in cytoplasm. Upon H2O2 exposure, the fluorescent signal from rhodamine 123 becomes diffuses, which was considered a diffused pattern. L-NAT pretreatment reversed the H2O2-induced dissipation of rhodamine123 fluorescence, indicating that ΔΨm was maintained (Figs. 6E–6F).

In addition to the rhodamine 123 staining pattern, the fluorescence intensity was also investigated. As shown in Fig. 6G, the fluorescence intensity of rhodamine 123 in H2O2 group decreased significantly compared to the untreated group, which implied the loss of ΔΨm, and L-NAT administration significantly inhibited the decreased fluorescence intensity induced by H2O2. Those data provide evidence that L-NAT pretreatment could prevent the dissipation of ΔΨm induced by H2O2.

Effect of L-NAT on mitophagy

To further explore whether the protective effect of L-NAT on hepatocytes was achieved by acting on mitophagy, we first measured the level of Beclin1 in H2O2-mediate BRL cells by immunofluorescence staining, then labeled mitochondrial by MitoTracker to observe the changes of mitophagy in H2O2-mediated BRL oxidative damage model. The data indicated that the number of co-localization of Beclin1 and MitoTracker were significantly increased after H2O2 induction. L-NAT effectively reduces H2O2-induced the number of Beclin1 and MitoTracker double positive cells induced by H2O2 (Fig. 7). This observation suggested that L-NAT could decreased the co-location of autophagy protein and mitochondrial.

Figure 7 Colocalization of Beclin1 and MitoTracker in H2O2-induced BRL oxidative damage model (200×).

Bar = 50 µm.

The autophagy inhibitor on cytoprotecyive effect of L-NAT

The above experiments have clarified that L-NAT can alleviate HIRI by inhibiting mitophagy. Subsequently, we investigated the effect of 3-MA, an autophagy inhibitor, on the hepatoprotective of L-NAT. According to preliminary experiment, the cell viability reached the highest point after 5 mM 3-MA pretreatment, and 5 mM 3-MA was subjected for subsequent experiments. As showed in Fig. 8A, L-NAT administration could elevate the decline of cell viability which was induced by H2O2 and 3-MA pretreatment could reverse part of those changes. Interestingly, the results of Western blotting and qRT-PCR revealed that the combination of 3-MA elevated the Beclin1 level of H2O2-induced BRL cells, compared with the usage of L-NAT solely (Figs. 8B–8D), implied that the autophagy inhibitor 3-MA partially weaken the protective effect of L-NAT on HIRI. Taken together, these results futher identified that the hepatoprotective effect of L-NAT depended on mitophagy process.

Figure 8 Effects of 3-MA on cytoprotection of L-NAT.

(A) Effects of different concentrations of autophagy inhibitors on cytoprotection of L-NAT (n = 3). (B) The mRNA levels of Beclin1 in BRL cells (n = 3). (C) Western blotting analysis of Beclin1 protein expression in BRL cells (n = 3). The data are the mean ± SD, *p < 0.05, **p < 0.01, ***p < 0.001 compared with the related group, #p < 0.05, ##p < 0.01, ##p < 0.001 compared with the related H2O2 group, ▴▴p < 0.01 compared with the related L-NAT group.

Discussion

Although the mechanism of HIRI has not been clarified, increasing evidence shows that oxidative stress caused by overproduction of ROS and depletion of endogenous antioxidants play a major role in contributing to tissue injury and thus liver dysfunction after HIRI. L-NAT, an antagonist of NK-1R, has the effect of scavenging ROS. In the present study, we demonstrated the following findings: first, we found that the changes of morphology and cell viability were improved by L-NAT. Second, L-NAT could significantly decrease autophagy after HIRI and resue the damage of mitochondrial structure and function. Finally, the hepatoprotecive effect of L-NAT was reversed by autophagy inhibitor. These findings implied that the protective effect of L-NAT on hepatocytes was closely related to inhibiting the excessive mitophagy indunced by HIRI.

Autophagy is a highly conservative intracellular process involving the degrading the impaired proteins or damaged organelles to maintain cellular homeostasis and supply substrates for energy generation, implying modulate autophagy could improve the ability of cells to stimulation. Therefore, autophagy is recognized as a critical pathway in the regulation of cell death and survival (Baehrecke, 2005; Hou et al., 2019). Although autophagy has been investigated more and more extensively in the field of liver diseases, the exact function of autophagy, which can be destructive or protective in HIRI remains controversial.

Studies demonstrate that autophagy plays important roles in protecting against HIRI. Zhao et al. (2019) reported that ulinastatin led to an increase in Beclin1, LC3-II and a decrease in P62. And 3-MA, an inhibitor of autophagy, could reverse the changes of the autophagy related proteins, and made morphological damage and cell apoptosis worsen in ulinastatin-treated H / R liver cells. Lee et al. (2016) showed that everolimus protected the liver against hepatic IRI by way of activating autophagy and the blockage of autophagy, which was abrogated by either bafilomycin A1 or si-autophagy-related protein 5. Rao et al. (2017)’s study indicated that isoflurane could reduce liver injury and restore liver autophagy by elevated LC3-II protein levels accompanied with increased P62 degradation. In addition, 3-MA pretreatment showed no significant influence in the control group, but abrogated the protective role of isoflurane preconditioning both in stressed livers. Furthermore, the beneficial effects of autophagy during I/R have been reported by other researchers (Wu et al., 2018). The aforementioned studies suggested that an upregulation of autophagy during I/R injury is protective. In the present study, we demonstrated that L-NAT pretreatment decreased the expression of LC3-II, Beclin1 and ATG-7 induced by HIRI, and increased the expression of P62, which were partly abolished by the combination of 3-MA. Those results implied that L-NAT likely protected hepatocyte by inibiting excessive autophagy production.

As the center of energy production, programmed cell death, reactive oxidative phosphorylation and calcium homeostasis, mitochondria plays an important role in regulating cell death and mitochondrial dysfunction is the key mechanism of various diseases including IR injury. As one type of selective autophagy, the major function of mitophagy is to identify and eliminate damaged or dysfunctional mitochondria. It is generally perceived that the function of autophagy is mainly prosurvival during I/R injury of various organs. Nevertheless, the true roles of activated mitophagy in I/R injury, such as myocardial I/R Injury (Yang et al., 2019), cerebral I/R (Li et al., 2018a; Li et al., 2018b; Li et al., 2018c; Li et al., 2018d), and renal I/R (Li et al., 2018a; Li et al., 2018b; Li et al., 2018c; Li et al., 2018d) etc., is still controversial. Mitophagy was reported to promote the clearance of injured mitochondria and then attenuate cell death induced by HIRI. Ning et al. (2018) demonstrated that parkin mediated mitochondrial autophagy was upregulated post-HIRI, leading to decreased hepatocyte death. However, parkin deficiency elevates HIRI by decreasing mitochondrial autophagy and increasing apoptosis. Bhogal et al. (2018) found that autophagy within LEC is reduced accompanying IRI increased cell death. Li et al. (2018a), Li et al. (2018b), Li et al., (2018c) and Li et al. (2018d) demonstrated that impairing mitophagy contributed to aggravation of hepatic ischemia-reperfusion injury in aging mice. Those data directly support the idea about the protective response of mitophagy against I/R injury.

However, excessive mitophagy and unnecessary mitochondrial clearance may sharply reduce the number of mitochondria and energy production, leading to cell death and strategies for suppressing mitophagy may protect cells from I/R injury (Lan et al., 2018). Li et al. (2018a), Li et al. (2018b), Li et al., (2018c) and Li et al. (2018d) demonstrated that aging aggravates hepatic ischemia-reperfusion injury in mice by impairing mitophagy. The degree of HIRI was aggravated by the enhancement of mitophagy and alleviated by parkin silencing impaired mitophagy. Sun et al. (2019) found that miR-330-3p could alleviate HIRI by inhibition PGAM5-induced mitophagy. In this study, we found that L-NAT not only resue the damage of mitochondrial structurecan, but also ameliorate the damage of ATP, mtDNA and loss of ΔΨm which induced by HIRI. Co-localization of the mitochondrial marker, MitoTracker, and Beclin1, the autophagosomal marker, was reduced by L-NAT undering H2O2-induced. Furthermore, the experiment in vitro also showed that autophagy inhibitor 3-MA intervening led to an increase of beclin1 expression in BRL cells exposed to H2O2 treatment. These findings further supported that the hepatoprotecive of L-NAT is achieved by inhibiting excessive mitophagy.

Taken together, these results suggested that hepatic ischemia and 6 h after reperfusion trigger excessive mitophagy, and L-NAT ameliorates I/R-induced hepatic injury through regulation of excessive mitophagy.

Conclusion

The present findings demonstrated that excessive mitophagy involved in the process of HIRI and L-NAT may protect hepatocytes against HIRI by inhibiting activation of excessive mitophagy and improving the structure and function of mitochondria.

Supplemental Information

Supplemental Information 1 qRT-PCR gene sequences

Click here for additional data file.

Data S1 Raw data exported from the hematoxylin-stained protein bands

Click here for additional data file.

Data S2 Raw data exported from the cropped Western Blot applied for data analyses

Click here for additional data file.

Supplemental Information 4 Western Blot

Click here for additional data file.

We are grateful to Zhijun Liu, who is an assistant professor and has worked in the University of Massachusetts Medical School for 8 years, for correcting English grammar, spelling and sentence structure throughout the manuscript.

Additional Information and Declarations

Competing Interests

Author Contributions

Animal Ethics

Data Availability

The authors declare there are no competing interests.

Huiting Li and Yitong Pan performed the experiments, authored or reviewed drafts of the paper, and approved the final draft.

Hongjuan Wu performed the experiments, prepared figures and/or tables, and approved the final draft.

Shuna Yu, Jianguo Li and Jiying Jiang conceived and designed the experiments, authored or reviewed drafts of the paper, and approved the final draft.

Jianxin Wang analyzed the data, prepared figures and/or tables, and approved the final draft.

Jie Zheng analyzed the data, authored or reviewed drafts of the paper, and approved the final draft.

Can Wang conceived and designed the experiments, prepared figures and/or tables, and approved the final draft.

The following information was supplied relating to ethical approvals (i.e., approving body and any reference numbers):

Weifang Medical University Medical Ethics Committee provided full approval for this research (No. 2017253).

The following information was supplied regarding data availability:

The raw measurements are available in the Supplemental Files.

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
