# Peer review of "Inhibition of excessive mitophagy by N-acetyl-L-tryptophan confers hepatoprotection against Ischemia-Reperfusion injury in rats"

_PeerJ, doi:10.7717/peerj.8665_

## Round 0.1 · original submission · Major Revisions

Dear Dr. Jiang,

Thank you for submitting your manuscript " Inhibition of excessive mitophagy by N-acetyl-L-tryptophan confers hepatoprotection against Ischemia-Reperfusion Injury" to PeerJ. Your manuscript has been reviewed by the Editors and three outside referees (see comments below). Although the topic is certainly of interest to the journal, serious technical and conceptual concerns voiced by the referees preclude publication of the current version of the manuscript in PeerJ.
Papers accepted for publication must meet methodological soundness, of high scientific quality. Unfortunately, it is the opinion of the expert reviewers that your submission has not yet met the criteria for acceptance at this stage of submission to PeerJ.

We regret that we could not render a more favorable decision on this version, but we appreciate the opportunity to re-review your work. We look forward to receiving your revision based on reviewers comments for our consideration.

Thank You
Sarwat Naz, PhD
Academic Editor

Reviewer 1 ·

Basic reporting

Literature background, figure structure and hypothesis are well established. English proficiency needs work.

Experimental design

Research question and experimental methodology well defined.

Validity of the findings

Conclusions and the data are well presented, needs few things to elaborate (see comments).

Additional comments

Comments on manuscript # 41725 " Inhibition of excessive mitophagy by N-acetyl-L-tryptophan confers hepatoprotection against Ischemia-Reperfusion Injury" by Li et. al.

The author studied the role of N-acetyl-L-tryptophan in hepatoprotection by inhibiting mitophagy against Ischemia-reperfusion injury in vitro and in-vivo.

Following things need to be addressed in the MS.
1. Line 186-200- Author studied the autophagy markers in rat liver tissues after I/R or I/R+L-NAT Treatment-These are all nice studies. But, what is missing how different is I/R rat compared to L-NAT treated ones in terms of metabolism, body weight, whole liver morphology or pathology or even other symptoms, like energy loss in terms of movement for food or water or balance on a beam from one end to another. The animal study needs to be elaborated or explained further in MS.

2. Figure 2: H&E staining should be included to see the proliferation/tissue integrity of the tissues used for Immunofluorescence staining?

3. Proofreading is missing all over the MS, e.g. line 116 and 150, Where is Dark? Should be in dark. Line 148: Living cells-should be live cells. Line 220-221: improve mitochondrial disfunction –that makes reverse of the result-should be: rescue mitochondrial dysfunction.

The paper needs nice English proficiency. Comments above- will make the MS better.

·

Basic reporting

1. All acronyms should be defined (expanded form in parenthesis) whenever mentioned for the first time in the text, including abstract.
2. English grammar and sentence structure need significant attention throughout the manuscript.
3. Some citations are either missing or mentioned very late in the text, for example, line 55-54 and 67-69. Also, Li et al 2013 (line 60) is missing from the reference section.
4. Figure 1B and C- relative expression of beclin 1 compared to the housekeeping gene GAPDH in Fig 1C at 4 hours is higher than 1hr. However, in the Weston blot image that does not appear to be the case. Similarly, the ratio for 24 hours seems lower than 6 hours, but in the western blot image it appears to be same as 6 hours. This discrepancy should be resolved. Additionally, there are no error bars on Sham group. It is important to mention where biological replicates for the sham group were included in the experiment or not, to account for baseline variation.
5. There is no error bar in sham group in any of the graphs. Does that mean no biological/experimental replicates were used?
6. In figure 1A, there seem to be a marginal increase in the relative mRNA expression however, there is a two-fold induction in the same in figure 2A. It is important to address the inconsistency between the experiments. Also, the timeline that was used as end point in Figure 2 should be mentioned in the figure legend.
7. In Figure 2 J, last panel, P62 is unusually high for sham control. What does the author think could be the reason? Similar observation for Figure 3J. It is important to include IgG controls in the panel for both Figure 2J and 3J to account for non-specific staining.
8. In figure 4 and 5, quantitation of autophagosomes and damaged mitochondria (if there are multiple image fields) will be more informative than providing single images for different conditions.
9. In line 221, the authors mistakenly mention mitochondrial function as dysfunction as it is implied that L-NAT can improve mitochondrial function.
10. Figure 7- a quantitative graph for the percent colocalization of beclin 1 with the mito-tracker dye should be included.
11. In Figure 8 the cell viability in L-NAT treated group does not seem to reach to sham level and is only marginally higher than the H2O2 group. Can the author speculate the reason behind it in context with protective effect of L-NAT. What dose of 3-MA was used to compare for beclin-1 expression in the mRNA and protein quantitation?

Experimental design

Biological/experimental replicates are missing from the control samples.

Validity of the findings

Some findings, as outlined in the basic reporting section, need to be re-evaluated carefully.

Reviewer 3 ·

Basic reporting

No comment.

Experimental design

No comment.

Validity of the findings

No comment.

Additional comments

The paper entitled “Inhibition of Excessive Mitophagy by N-acetyl-L-tryptophan Confers Hepatoprotection against Ischemia-Reperfusion Injury” by Huiting Li et al investigated the effects of L-NAT in HIRI models both in vitro (BRL cells) and in vivo (Sprague-Dawley (SD) rats). The authors concluded that L-NAT could protect hepatocytes against HIRI by inhibiting excessive mitophagy.

Overall, conclusions are well stated and linked to original research question. Literature is relevant and well referenced.

However, the following concerns need to be addressed:

The authors claim that BRL cells and rats are under I/R injury after H202 or vessel clamp surgery treatment, and literature indicates that autophagy correlates with I/R injury. While the authors clearly showed the marker proteins for autophagy in all experiments, it is unclear to me what is the evaluation criteria for I/R injury. Are there markers to indicate that the treated rat or cells are correct I/R models? What are the evaluation factors to assess that rat and BRL cells are really injured after respective treatments?

The author concluded that L-NAT could protect hepatocytes against HIRI by inhibiting excessive mitophagy. Please explain differences between “normal mitophagy” and “excessive mitophagy” How did the authors differentiate between the two during the rat and cellular experiments?

Are all proteins in figure 2E from the same gel (membrane) but with different contrasts? It seems that P62 and GAPDH are from a different gel.

Please include more descriptions for the immunofluorescence result in Figure 2J. How did authors quantify the fold changes between I/R and I/R-LNAT group? Please also use other color (eg. white) for “DAPI” characters on top.

The autophagosomes in figure 4 and the mitochondria in figure 5 are quite similar in shape and morphology. Since no specific antibody was used in transmission electron microscopy, the author should specifically explain how mitochondria and autophagosomes are differentiated under TEM. Otherwise specific bio-markers for mitochondria and autophagosome need to be used to distinguish between the two.

Please include more detailed description for Figure 4 and 5, including how did the volume changes calculated?

In I/R group of Figure 5, it looks like the majority mitochondria still have normal shape. Please explain.


The introduction needs more detail. I suggest that authors improve the description at lines 62 to provide more justification for how “mitophagy may be a new therapeutic target for HIRI”.

The English language should be improved to ensure that an international audience can clearly understand your text. Corrections including but not limited to:

Line 67, “And another study also shown that L-NAT, an antagonist of the neurokinin 1 receptor (NK-1R) known to disrupt the binding of Substance P (SP) to NK-1R, significantly protective effect on neurodegenerative diseases.” may change to “Another study also shown that L-NAT, an antagonist of the neurokinin 1 receptor (NK-1R) known to disrupt the binding of Substance P (SP) to NK-1R, shows significantly protective effect on neurodegenerative diseases.”

Line 71, “but also alleviate the mice model of ALS and cerebral ischemia / hypoxia injury in vivo” change to “but also alleviate ALS and cerebral ischemia / hypoxia injury in mouse model in vivo ”

Line 73, “Recent work confirmed that hepatoprotective effects of…” change to “recent work confirmed the hepatoprotective effects of…””


Minor corrections:

Put more space between 2E and 2F, 3E and 3F

---

## Round 0.2 · accepted · Accept

Dear Dr. Jiang,

Thank you for your submisiion of the revised manuscript titled " Inhibition of excessive mitophagy by N-acetyl-L-tryptophan confers hepatoprotection against Ischemia-Reperfusion Injury in rats " to PeerJ.
I am writing to inform you that your revised manuscript has been re-reviewed by two reviewers who found all the incorporated changes satisfactory and helpful in making the quality of the work Acceptable for publication. Congratulations!

·

Basic reporting

The authors have made significant revisions in the English language and have made most of the suggested changes in the manuscript. Therefore, at this point, the manuscript can be accepted for publication.

Experimental design

Satisfactory

Validity of the findings

satisfactory

Reviewer 3 ·

Basic reporting

no comment

Experimental design

no comment

Validity of the findings

no comment

Additional comments

The author has answered all questions raised.